# DiffraNet: Automatic Classification of Serial Crystallography Diffraction Patterns

## Abstract

Serial crystallography is the field of science that studies the structure and properties of crystals via diffraction patterns. In this paper, we introduce a new serial crystallography dataset comprised of real and synthetic images; the synthetic images are generated through the use of a simulator that is both scalable and accurate. The resulting dataset is called DiffraNet, and it is composed of 25,457 512x512 grayscale labeled images. We explore several computer vision approaches for classification on DiffraNet such as standard feature extraction algorithms associated with Random Forests and Support Vector Machines but also an end-to-end CNN topology dubbed DeepFreak tailored to work on this new dataset. All implementations are publicly available and have been fine-tuned using off-the-shelf AutoML optimization tools for a fair comparison. Our best model achieves 98.5% accuracy on synthetic images and 94.51% accuracy on real images. We believe that the DiffraNet dataset and its classification methods will have in the long term a positive impact in accelerating discoveries in many disciplines, including chemistry, geology, biology, materials science, metallurgy, and physics.

## 1 Introduction

Real-time feedback on diffraction images is vital in Crystallography (Berntson et al. (2003); Ke et al. (2018)). Crystallography (Woolfson (1997)) is the science that studies properties of crystals. It makes use of X-ray diffraction to infer structures of crystals. Broadly, a crystal is irradiated with an X-ray beam that strikes the crystal and produces an image with the diffraction pattern (Fig. 1). Images are captured by a detector that runs at 130 Hz. At present serial crystallography, scientists have to screen tons of images by manual classification. This process is not only error-prone but also has the effect of slowing down the overall discovery process.

In this paper, we introduce a method for generating labeled diffraction images. The technique produces and labels images via a simulator and, therefore, the process is both scalable and accurate. The simulator receives as input the properties of the incident X-ray beam, the environment, and the structure to be analyzed and generates synthetic diffraction images. Since the process is simulated and controlled, the dataset annotation is 100% accurate, an impossible feat for manually annotated real images.

As a result of the simulator we introduce DiffraNet, the first dataset of serial crystallography diffraction that combines real and synthetic images. DiffraNet is composed of 25,457 512x512 grayscale labeled images and we open it to the rest of the community. The synthetic images are divided into five classes, each representing a possible outcome of the serial crystallography experiment. Of the five possible classes, two classes denote images with no diffraction patterns (an undesired outcome) and the other three denote images with varying degrees of diffraction. The real images are divided into two classes, representing images with and without diffraction patterns. DiffraNet contains fundamentally different images with respect to standard image datasets such as ImageNet (Deng et al. (2009)) and the CIFARs (Krizhevsky (1993)), see Fig. 2.

Finally, we also present three different approaches for classifying diffraction images. First, a method based on a mix of feature extractors and Random Forests (RF). Second, a combination of feature extractors and Support Vector Machines (SVM). Last, DeepFreak, a Convolutional Neural Network (CNN) topology based on ResNet-50. All approaches are open-source and have been fine-tuned using AutoML hyperparameter optimization tools such as Hyperopt (Bergstra et al. (2013)) and

BOHB (Falkner et al. (2018)). These approaches achieve 98.45%, 97.66%, 98.5% accuracy on synthetic data, respectively, and 86.81%, 91.1%, 94.51% accuracy on real data.

To sum up, the contributions of this paper are:

- A new, openly available, classification dataset dubbed *DiffraNet* for image diffraction in the serial crystallography experimental setting.
- Three classification methods based on RFs, SVMs and CNNs able to classify synthetic diffraction images with up to 98.5% accuracy and real diffraction images with up to 94.5% accuracy.
- An open-source implementation of the newly introduced approaches.

The rest of this paper is organized as follows. Section 2 presents a background in X-ray crystallography and a summary of related work. Section 3 describes our simulator and the DiffraNet dataset. Sections 4 and 5 describe our approaches for classifying diffraction images from DiffraNet and the experimental results achieved. Finally, Section 6 concludes this work and presents future work.

## 2 BACKGROUND

### 2.1 CRYSTALLOGRAPHY

Crystallography is used in many disciplines, including chemistry, geology, biology, materials science, metallurgy, and physics. It has been a central tool in driving significant increases in understanding processes from solid-state physics to molecular biology to synthetic chemistry (Woolfson (1997)). This understanding, in turn, has led to substantial advances in, for instance, drugs development for fighting diseases. Serial Crystallography (Stellato et al. (2014)) refers to a more recent crystallography technique for investigating properties from hundreds of thousands of microcrystals using X-ray free-electron laser.

Crystallography makes use of X-ray diffraction to infer the structure of crystalline samples. First, a crystal is irradiated with an X-ray beam. As X-ray photons strike the crystal, some will diffract due to the geometry of the lattice and produce a diffraction pattern unique to the material as in Fig. 1. These patterns are recorded by a detector (usually phosphor or silicon) and make it possible to infer information about the crystal, like the chemical bonds and disorder of its atoms.

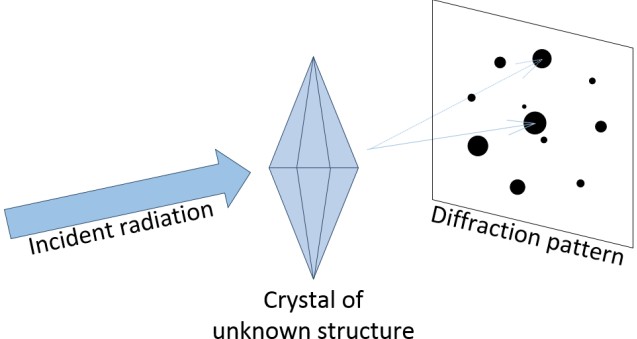

Figure 1: Generic scheme depicting a crystallography experiment.

Analysis and feedback on diffraction images are paramount in both conventional and serial crystallography (Berntson et al. (2003); Ke et al. (2018)). Recent technological advances have automated and accelerated crystallography experiment steps and, in turn, allowed researchers to generate diffraction results at unprecedented speeds. However, as no system currently exists to provide real-time analysis of the diffraction images produced, many of the compelling advantages afforded by these technological leaps cannot be fully utilized. Besides, without timely feedback, expensive and limited quantity samples may be wasted because of problems regarding experimental optimization, sample positioning, or X-ray beam alignment. This paper addresses the automation of serial crystallography image screening.

## 2.2 PREVIOUS APPROACHES TO CLASSIFICATION ON SERIAL CRYSTALLOGRAPHY

Several studies have been done for trying to automatically classify images derived from crystallography phenomena (Berntson et al. (2003); Becker & Streit (2014); Yann & Tang (2016); Park et al. (2017); Bruno et al. (2018); Ke et al. (2018); Ziletti et al. (2018)).

In particular, Bruno et al. (2018) employed CNNs for classifying outcomes of crystallization processes. The model they used is a variation of *Inception-v3* (Szegedy et al. (2016)), images were categorized in the following four classes: clear, precipitate, crystal, and other. The dataset used in this study has nearly half a million images, and around 10% of them were used for testing. They achieved 94% accuracy on the test set, approximately. Yann & Tang (2016) aimed at analyzing protein crystallization-trial images. Notably, their CNN approach dubbed *CrystalNet* hits around 8% and 20% improvement in overall accuracy compared to the Random Forests and Nearest Neighbor approaches, respectively.

Ziletti et al. (2018) used CNNs to classify crystal structures, i.e., the way atoms inside a crystal are arranged. By using diffraction images, they were able to represent and classify a dataset with around 100,000 crystal structures. Park et al. (2017) worked on classifying powder X-ray diffraction patterns using CNNs achieving 94.99% of accuracy.

Similar to our work, Ke et al. (2018) used a CNN for detecting Bragg spots on crystallography diffraction images. Their CNN employs a structure similar to that of *AlexNet* (Krizhevsky et al. (2012)) and comprises four sets of layers: convolution, batch normalization, rectification, and downsampling (max pooling). They used local contrast normalization to enhance the contrast between background and Bragg spots. They also augmented the dataset through the use of random and center cropping. Ke *et al.* used a human expert annotated dataset, consisting of 2,000 images, as the ground truth and compared their CNN accuracy against with automatic spot-finding tools. They achieved around 93% accuracy in classifying images as a *hit*, *maybe*, or *miss*. Respectively, these classes refer to when an image does, might, and does not possess Bragg spots.

Our work sets apart from these above in several ways. First, our process of labeling data is scalable and accurate. Second, our dataset is tailored to a specialized application: Crystallography. Third, we explore different computer vision techniques for classification. Last, we use multiple AutoML optimization tools to achieve the best results in each setting.

## 2.3 IMAGE DATASETS

Today, there is a great deal of publicly available datasets for training machine learning models. Few notorious datasets are: *ImageNet* (Deng et al. (2009)), *CIFAR-10/100* (Krizhevsky (1993)) and *COCO* (Lin et al. (2014)). ImageNet, for example, comprises around 14 million images following the WordNet hierarchy organization. On average, each node of the ImageNet's hierarchy has 500 images. Some popular ImageNet synsets include *animal, plant, material, and activity*. Common Objects in Context, or COCO for short, is an annotated dataset consisting of images portraying scenes from everyday life and their ordinary objects. COCO features, for instance, 200,000 labeled images, 1.5 million object instances, and 250,000 people with keypoints. The CIFAR-10 and CIFAR-100 are annotated samples of the Tiny Images Dataset (Krizhevsky (1993)). CIFAR-10 comprises 60,000 images divided into ten (airplane, automobile, bird, cat, deer, dog, frog, horse, ship, truck) classes containing 6,000 images each. Of these, 50,000 are for training and the rest for testing. Its larger counterpart, CIFAR-100, is much like CIFAR-10, but it is made up of 100 classes with 600 images each.

## 3 THE DIFFRANET DATASET

We introduce a new, openly available, dataset of diffraction images dubbed *DiffraNet*. DiffraNet is comprised of both real and synthetic diffraction images. However, experimental diffraction images are difficult to classify on a large scale. Highly trained experts are needed to categorize these images manually, and the process is both slow and error-prone. Thus, the majority of our dataset is synthetically generated. The synthetic dataset is 100% accurate because labels derive images, not the other way round.

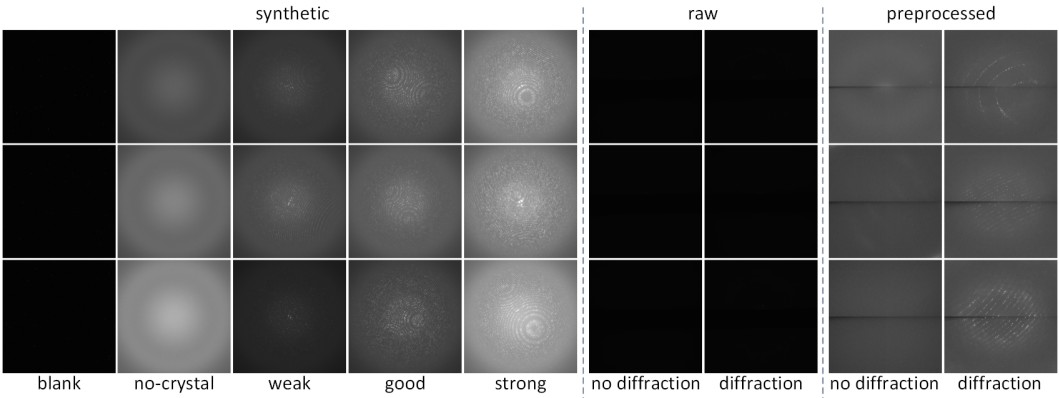

Figure 2: DiffraNet dataset synthetic and real classes.

Our synthetic images were generated using the nanoBragg simulator[1]. The physics of X-ray diffraction are well understood and we have developed simulators for the entire process of producing diffraction images. The input to nanoBragg includes X-ray beam properties (flux, beam size, divergence, and bandpass), crystal properties (unit cell, number of cells, and a structure factor table), and the experimental parameters (sources of background noise, detector point-spread, and shadows – such as the beamstop). The simulation is computationally intensive but highly parallelizable: images can be rendered and labeled at an average rate of one per second on the 384-core SMB cluster.

The images were generated using a single crystal structure, but different diffraction parameters. Most notably, the X-ray beam intensity varied widely from shot to shot, as did the volume of crystalline material in that beam relative to non-crystalline matter. This wide dynamic range is a big factor in making this kind of data difficult to analyze. We also simulate imperfections in the crystal by breaking it up into smaller crystals and vary parameters like the sources of background noise and the orientation of the crystal. At last, we convert the 16-bit images generated by the simulator to 8-bits by taking the square root of each pixel. For a detailed description of the entire simulation process, the reader can refer to Appendix A.

DiffraNet comprises 25,000 512x512 grayscale synthetic diffraction images. The classes are *blank*, *no-crystal*, *weak*, *good*, or *strong*. *Blank* denotes an image with no X-rays and only detector noise while *No-crystal* the diffraction from amorphous carrier material but no crystalline matter. *Weak*, *Good*, and *Strong*, in turn, denote images with a crystal in the beam with increasingly stronger contribution to the pattern: *Weak* has small or faint diffraction patterns, *Good* has slightly larger and more discernible patterns, and *Strong* are ideal images, with large and clear diffraction patterns.

DiffraNet also comprises 457 512x512 grayscale real diffraction images. Real images have higher resolution, are notably darker, and include a horizontal beamstop shadow across the middle that blocks part of the diffracted beams. We downsample and crop these images down to 512x512 resolution, removing the beamstop shadow, and provide two real dataset variants in DiffraNet: one with the raw cropped images and another with the images preprocessed to make the patterns more visible. The preprocessed images were generated by multiplying the pixels of the raw images by a constant factor so that their mean pixel value matches the mean pixel value of the synthetic images. Finally, because accurately labeling real images is a challenging and expensive task, we label these images simply as *diffraction* and *no-diffraction*. Fig. 2 shows samples from each class of DiffraNet's synthetic and real datasets.

DiffraNet is publicly available[2] and can be used for training, validating, and testing machine learning models. The primary goal in the classification of DiffraNet is to differentiate between classes with and without crystal diffraction patterns so that images without diffraction pattern can be discarded and downstream analysis can focus on images that are the most promising. DiffraNet partitions the synthetic dataset into training (40% of the dataset with a total of 10,000 images), validation (9.6%

---

[1]http://doubleblind.com
[2]http://doubleblind.com

of the dataset with a total of 2,400 images), and test sets (50.4% of the dataset with a total of 12,600 images) and the real dataset into validation (~80% of the dataset with a total of 366 images) and test (~20% of the dataset with a total of 91 images) sets.

## 4    CLASSIFICATION ON IMAGE DIFFRACTION

We propose three approaches for the classification of the DiffraNet dataset introduced in 3. The first two approaches rely on RFs and SVMs combined with feature extractors and the third approach is a CNN. We use off-the-shelf AutoML tools to search the hyperparameter space of the three classifiers automatically. We adopt two different tools: Hyperopt (Bergstra et al. (2013)) for the RF and SVM classifiers, and BOHB (Falkner et al. (2018)) for the CNN classifier. We use BOHB for the CNN because it includes a multi-fidelity feature that accelerates searches on CNNs. Both tools are based on Tree Parzen Estimator (TPE) (Bergstra et al. (2011)) models.

### 4.1    FEATURE EXTRACTORS

We implement three feature extractors to use together with our RF and SVM classifiers. Specifically, we use the Scale Invariant Feature Transform (SIFT, Lowe (2004)) with the Bag-of-Visual-Words approach (BoVW, Yang et al. (2007)) as local feature extractor, and the Gray-level Co-occurrence Matrix (GLCM, Haralick et al. (1973)) and Local Binary Patterns (LBP, Ojala et al. (2002)) as global feature extractors. We choose these extractors because of their strong performance in image classification tasks (Kumar et al. (2017)) and in particular GLCM and LBP for their global texture features that are suitable in describing the images in DiffraNet.

We implement the feature extractors in Python using the OpenCV and scikit-image libraries. Also, we fine-tune the parameters of the extractors and both the SVM and RF classifiers using the Hyperopt Python library (Bergstra et al. (2013)). For the SIFT + BoVW extractor, we use the k-means algorithm to aggregate the visual codewords and optimize the size of the codebook by tuning the number of clusters in the k-means algorithm. For GLCM, we tune the distances and angles between pixel value pairs and use six Haralick features (Haralick et al. (1973)): Contrast, Dissimilarity, Homogeneity, Angular Second Moment, Energy, and Correlation. Finally, for LBP we tune the radius and number of points parameters that define the neighborhood size used by the extractor to compute the binary patterns. The feature extractors search space is summarized in Appendix B.

### 4.2    RF AND SVM CLASSIFIERS

RFs (Breiman (2001); Criminisi et al. (2012)) is an ensemble learning technique that can be used for both classification and regression. RFs create a forest of *decision trees*, a supervised learning technique for decision-making processes. A *randomized decision tree*, in turn, randomly select attributes out of a set of randomly chosen training samples.

SVMs (Vapnik (1995)) is a supervised learning technique used for classification and regression. For (almost) linearly separable data SVMs are straightforward: given labeled training data, SVMs output a separating hyperplane which may be then used to classify unlabeled data. For data that is not linearly separable, on the other hand, SVMs first employ *kernel functions* to map that onto another—often higher—dimensional space where the data is (almost) linearly separable and then, accordingly, proceed by finding a hyperplane.

We use Hyperopt to search for feature extractors and classifier hyperparameters jointly. Hyperopt proceeds by choosing one feature extractor and one classifier and then choosing a hyperparameter configuration based on the search spaces of each. By optimizing the extractor and classifier together, we allow Hyperopt to estimate particular extractor and classifier combinations that function well together. The search space is summarized in Appendix B.

### 4.3    THE DEEPFREAK NEURAL NETWORK

We introduce a new CNN dubbed *DeepFreak*. DeepFreak uses an adapted version of the Residual Neural Network with 50 layers (ResNet-50, He et al. (2016)). ResNet introduces *identity shortcut connections* that bypass one or more layers (as in Highway Networks, Srivastava et al. (2015)) with

Table 1: DeepFreak topology hyperparameters.

| Hyperparameter | Value |
|---|---|
| Number of filters | 64 |
| 1st convolution kernel | 7 |
| 1st convolution stride | 2 |
| 1st pool size | 3 |
| 1st pool stride | 2 |
| 2nd pool size | 7 |
| 2nd pool stride | 1 |
| Number of blocks | 3, 4, 6, 3 |
| Block strides | 1, 2, 3, 3 |

Table 2: DeepFreak learning hyperparameters.

| Hyperparameter | Value |
|---|---|
| Number of epochs | 180 |
| Optimizer | SGD |
| Learning rate | $8.4474 \times 10^{-4}$ |
| Decay epochs | Every 10 epochs |
| Decay rate | 0.1 |
| Momentum | 0.56168 |
| Weight decay | $3.4855 \times 10^{-5}$ |
| Loss function | Cross-Entropy |
| Batch size | 1 |

the addition of residual blocks which let the stacked layers fit a residual mapping instead of directly fitting the desired underlying mapping. This helps to address the vanishing gradient problem of deep networks.

The original ResNet topology, however, presumes images of size 224x224 as opposed to our 512x512 DiffraNet images. Further, we have found that simply downsampling our images to the image size accepted by ResNet leads to poor performance (96.79% training accuracy and 72.08% validation accuracy). Instead, we design a set of potential adjustments to ResNet's topology to intensify the network's downsampling while still enabling it to leverage additional information from our high-resolution images. We use PyTorch's official implementation of ResNet-50 as a baseline (PyTorch (2018)), implement our topology adaptations, and use BOHB (Falkner et al. (2018)) to find the best topology and hyperparameter combination for DeepFreak; the reader can refer to Appendix C for more details on the DeepFreak search space.

BOHB is an AutoML tool based on Hyperband (Li et al. (2017)) and Bayesian Optimization (Bergstra et al. (2011)). It uses an iterative algorithm parameterized by two hyperparameters: maximum budget and $\eta$. These hyperparameters define how many configurations are evaluated per iteration and for how many epochs the network uses each configuration. In every iteration, BOHB assigns a budget—equal to or lower than the maximum budget—to all the configurations sampled. For each iteration $i$, BOHB keeps $1/\eta$ of the configurations tested in the iteration $i - 1$ and increase the budget assigned to each configuration, up to the maximum budget. Our ultimate goal in using BOHB is to downsample the network so that DeepFreak trains faster and achieves higher accuracy.

We run BOHB on DeepFreak with a maximum budget of 50 epochs and $\eta = 3$. The best topology and hyperparameters found extends the strides of ResNet-50's last two blocks to 3 (instead of 2), uses a batch size of 1, a weight decay of $3.4855 \times 10^{-5}$, and a momentum of 0.56168. The learning rate starts at $8.4474 \times 10^{-4}$ and decays by 10 every 10 epochs. We split DiffraNet's synthetic dataset into training, validation, and testing (c.f. Section 3) and train the network for over 180 epochs. For each image in the training set, we rescale the pixel values to the [0, 1] range and subtract the per-pixel mean. DeepFreak configuration is summarized in Tables 1 and 2 and our code has been made publicly available (*authors omitted* (2018)).

## 5 EXPERIMENTS

In this section, we present the results of our hyperparameters search with Hyperopt and BOHB, as well as the performance of our models in the classification of DiffraNet. We first show results for synthetic images only and then evaluate our models on real diffraction images.

### 5.1 HYPEROPT RESULTS

We present the best configurations found by Hyperopt for the SVM and RF classifiers and their performance in the validation set. For this experiment, we have run Hyperopt for 150 iterations on a machine with 2 Intel Xeon E5 processors. The optimization has taken roughly 36 hours to complete, and the results are shown in Table 3.

Table 3: Best configuration of RF (left) and SVM (right) and accuracy on the validation set.

| Hyperparameter | Values |
|---|---|
| GLCM Distances | [1, 2, 5, 8] |
| GLCM Angles | [45, 135] |
| Max Depth | 20 |
| Max Features | $\sqrt{features}$ |
| Number of Trees | 100 |
| Class Weights | None |
| Accuracy | 98.58% |

| Hyperparameter | Values |
|---|---|
| GLCM Distances | [1, 5, 8] |
| GLCM Angles | [0, 90, 135] |
| C | 32 |
| $\gamma$ | 0.5 |
| Class Weights | [0.25, 0.25, 0.166, 0.166, 0.166] |
| Accuracy | 97.88% |

RF and SVM classifiers have achieved 98.58% and 97.88%, respectively, i.e., RF has performed slightly better than SVM (0.7%). Note that the highest accuracy for both SVM and RF use GLCM as a feature extractor. This accuracy indicates that the GLCM works better than LBP and SIFT in the DiffraNet dataset. Precisely, GLCM has been the best extractor (98.58% accuracy), with LBP closely behind (96.71% accuracy). These results corroborate our hypothesis that global texture extractors would fit DiffraNet better. On the other hand, the SIFT + BoVW extractor has achieved 56.4% accuracy. This low accuracy is not surprising since SIFT looks for features in corners and objects of images, which are unusual in images from DiffraNet. Table 4 exhibits the best results achieved by classifiers for each feature extractor.

Table 4: Feature extractors and models best accuracy on validation set.

| Feature Extractors | Model | Hyperparameters | Values | Validation Accuracy |
|---|---|---|---|---|
| GLCM | RF | Distances
Angles | [1, 2, 5 8]
[45, 135] | 98.58% |
| LBP | SVM | Points
Radius | 24
3 | 96.71% |
| SIFT | RF | Clusters | 25 | 56.42% |

## 5.2 BOHB RESULTS

We present the results of the DeepFreak optimization using BOHB. Here, we have run BOHB for 16 iterations in parallel in a machine with 2 Nvidia GeForce GTX 1080 Ti GPUs. The optimization has taken about nine days.

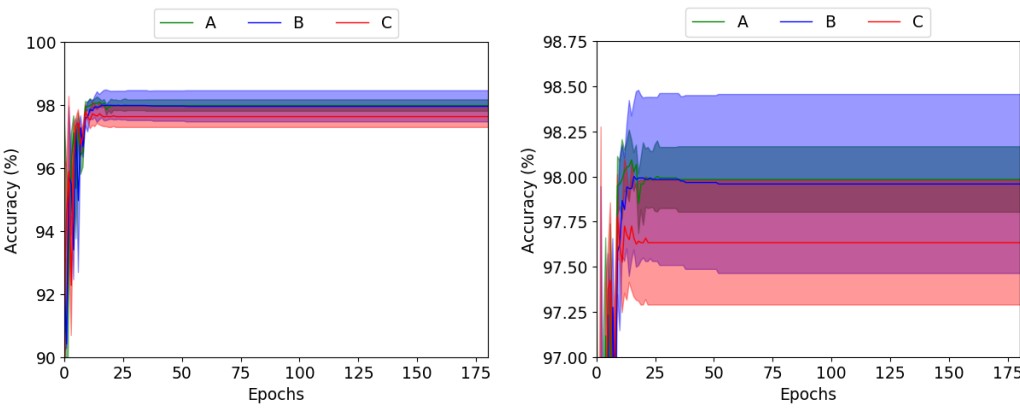

Figure 3: Mean accuracy (thin line) and 80% confidence interval (shade) of the three best configurations found by BOHB on the validation set. Each configuration was run five times for 180 epochs.

We have run the three best configurations (A, B, and C) found by BOHB, five times each. Figure 3 shows the mean learning curve (thin line) and the 80% confidence interval (shade) for each configuration. Note A and B have highest mean accuracy; B has a broader confidence interval. We believe A has a narrower confidence interval due to its larger pooling layer, which leads to faster downsampling. Given the variability of CNNs training, we use these curves to choose the best DeepFreak configuration by analyzing mean and variance of these configurations. We have chosen B due to its highest validation accuracy over the networks we trained. The details on the three configurations used in this experiment are shown in Appendix D.

The final learning curve for DeepFreak, with the best configuration found by BOHB, is shown in Figure 4; the details on this configuration have been discussed in Section 4.3. We have trained the network for 180 epochs; the training converged after around 20 epochs. After training, DeepFreak achieved 98.42% validation accuracy, a result similar to RF and superior to SVM.

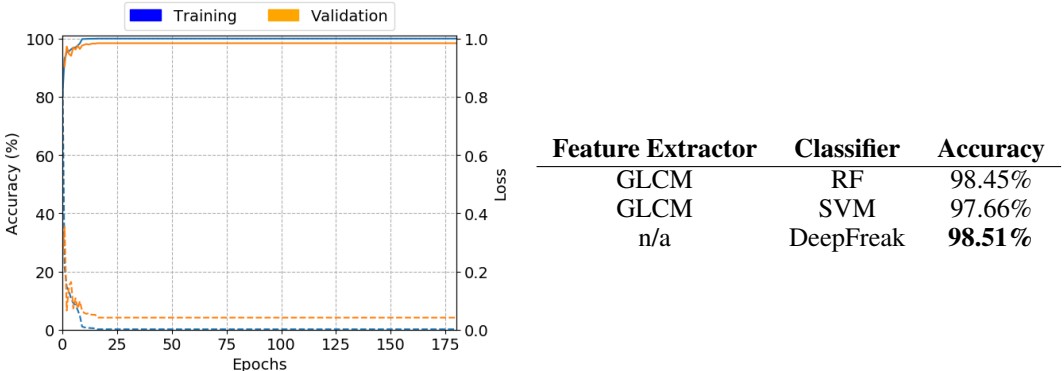

| Feature Extractor | Classifier | Accuracy |
|:---:|:---:|:---:|
| GLCM | RF | 98.45% |
| GLCM | SVM | 97.66% |
| n/a | DeepFreak | **98.51%** |

Figure 4: DeepFreak accuracy (solid line) and loss (dashed line) curves on training and validation sets; 180 epochs in total.

Table 5: DeepFreak, RF, and SVM accuracies on DiffraNet's test set.

## 5.3   TEST SET RESULTS

We have run the best configuration for our three classifiers over DiffraNet's test set, results are shown in Table 5. Note that the results are similar to those in the validation set, indicating that the three models can generalize the training data. Besides, all classifiers achieved over 97.6% accuracy. Notably, DeepFreak achieved the highest accuracy. Precisely, the accuracy of DeepFreak on the test set has been higher than that on the validation set, surpassing the RF and SVM by 0.06% and 0.85%, respectively.

We show the DeepFreak confusion matrix on the test set in Table 6; we show the RF and SVM confusion matrices on the test set in Appendix E. Note that misclassification often happens between *weak* and *good* and between *good* and *strong*. This behavior is natural since classes in these pairs are similar. Besides, note that DeepFreak hits near perfect results on the *blank* and *no-crystal* classes, as evidenced by their precision (99.95% and 99.45%) and recall values (100% and 99.94%). This result means we can discard images without diffraction patterns with 99.83% accuracy. This is an important result for this application domain because it is important not to discard useful images; as mentioned in Section 3, it is less problematic to misclassify between the classes *weak, good, strong*.

## 5.4   RESULTS ON REAL IMAGES

Last, we evaluate our models on DiffraNet real datasets. We first run our models from sections 5.1 and 5.2 on the real datasets to assess the impact of the reality gap. Table 7 shows the accuracy of each model on all 457 images from DiffraNet real datasets. We note that the reality gap degrades the accuracy of all of the models by at least 22.45%. DeepFreak was the least affected by the reality gap in both variants of the real dataset (22.45% and 26.46% accuracy loss). Conversely, RF was the most affected by the reality gap in both variants of the real dataset (54.56% and 53.22% accuracy

Table 6: DeepFreak confusion matrix for the test set.

|  |  | Predicted class | | | | | |
|---|---|---|---|---|---|---|---|
|  |  | blank | no-crystal | weak | good | strong | Recall (%) |
|  | blank | 2069 | 0 | 0 | 0 | 0 | 100 |
|  | no-crystal | 0 | 3266 | 2 | 0 | 0 | 99.94 |
| True class | weak | 1 | 18 | 3280 | 47 | 0 | 98.03 |
|  | good | 0 | 0 | 38 | 2368 | 38 | 96.89 |
|  | strong | 0 | 0 | 0 | 44 | 1428 | 97.01 |
|  | Precision (%) | 99.95 | 99.45 | 98.80 | 96.3 | 97.41 |  |

Table 7: Accuracy of our models on DiffraNet's real dataset before and after our AutoML optimization for real data.

| Pre-optimization | | | | Post-optimization | | | |
|---|---|---|---|---|---|---|---|
| Extractor | Classifier | Raw | Preprocessed | Extractor | Classifier | Raw | Preprocessed |
| GLCM | RF | 43.86% | 45.2% | LBP | RF | 90.11% | 86.81% |
| GLCM | SVM | 50.42% | 54.8% | LBP | SVM | 59.34% | 91.1% |
| n/a | DeepFreak | **76.06%** | **72.05%** | n/a | DeepFreak | **91.21%** | **94.51%** |

loss). These results indicate that, while our models perform well on the synthetic data, they do not generalize as well to the real data.

The results on Table 7 (left side) indicate that we have to improve the generalization of our models to real diffraction images. To do this, we repeat our AutoML optimization, this time, we optimize for performance on the real dataset. Namely, we split the real dataset into validation and test sets (c.f. Section 3) and use the AutoML tools to find the best configuration for each model based on the accuracy on the real validation set. We do not add real images to the training set, our goal is to find the models that generalize the best to real images, while training only on synthetic images. We show the best configuration found for each model and each dataset in Appendix F.

Table 7 (right side) shows the performance of the best configuration of each of our models on the real test sets. DeepFreak hits the highest accuracy on both real datasets (91.21% and 94.51% on raw and preprocessed datasets, respectively). Conversely, SVM (59.34%) and RF (86,81%) were the most affected by the reality gap on the raw and preprocessed real datasets, respectively. We note that all models have degraded accuracy on the real datasets, compared to the synthetic dataset. However, the high accuracy of our models shows that our simulated dataset can be effectively used to train models for real diffraction image classification.

## 6    CONCLUSIONS AND FUTURE WORK

We have tackled the challenge of real-time classification of serial crystallography diffraction images. We have developed a method for generating accurately labeled synthetic diffraction images and used that to generate DiffraNet. DiffraNet comprises 25,000 512x512 grayscale synthetic diffraction images, each tagged as one out of five classes representing possible outcomes from crystallography experiments. DiffraNet also comprises 457 512x512 grayscale real diffraction images, each tagged as one out of two classes representing desirable and undesirable outcomes from crystallography experiments. DiffraNet is publicly available and can be used for training, validating, and testing machine learning models tailored to crystallography.

We have also explored several computer vision classification approaches. They are based on a blend of standard feature extractors with the RF and SVM classifiers and on an end-to-end CNN architecture called DeepFreak. All of our approaches have been fine-tuned with AutoML tools and tested over DiffraNet. Our results show that DeepFreak obtained the highest accuracy on both synthetic and real diffraction images (98.51% and 94.51%, respectively). Moreover, DeepFreak achieved 99.83% accuracy in distinguishing between images with and without diffraction patterns.

In future iterations of the DiffraNet dataset we plan to add new images and new classes that are common place in serial crystallography. As an example, a class that is valuable in practice is to

detect the presence of ice in the images. Images with ice indicate problems with the experiment setup that can disrupt the results and even damage the detector. It is important to detect and address these problems in a real-time feedback loop.

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

## A    DiffraNet Simulation Procedure

The exact procedure used to generate DiffraNet is described here. First, the atomic structure of photosystem II (Protein Data Bank ID: 4rvy, doi: 10.2210/pdb4RVY/pdb) was downloaded from the Protein Data Bank (Berman et al. (2002)). The deposited coordinate files in the PDB do not explicitly contain a representation for the disordered solvent that floods the gaps between protein molecules inside the crystal, and neglecting this material in the diffraction pattern calculation leads to unrealistically strong spots at low deflection angles. The disordered solvent was therefore modeled as described by Tronrud (1997) using the program phenix.fmodel (Adams et al. (2010)) to create a list of "structure factors": F(h).

Structure factors are the coefficients of a Fourier transform of the map of electron density within a single unit cell of the crystal. These coefficients form a 3D array of floating-point values indexed by the 3-vector "h". It's integer-value components (h,k,l) are called the "Miller indices" in crystallography (Miller (1839)) and each corresponds to a potential x-ray spot on the detector. The definition of a structure factor (Hartree (1923)) is the ratio between the amplitude of the wave of light scattered by an object of interest to that scattered by a single electron located at the origin. In this case the "object of interest" is the unit cell of the photosystem II crystal. The absolute intensity of the spot on the detector is then obtained by multiplying this unit-cell structure factor by the classical Thomson scattering of a single electron and by the structure factor of the crystal lattice itself, which is obtained from the classic Fraunhofer grating equation described by Kirian et al. (2010). The intensity at a given pixel on the detector is proportional to the square of the structure factor, and directly proportional to the incident x-ray beam intensity.

In this simulation, the incident X-ray beam was given a mean pulse fluence of 1e12 photons focused into a 30 micron wide square spot at the crystal position. This intensity varied from shot-to-shot with a Gaussian distribution and the RMS fluctuation of the X-ray pulses was made to be equal to the mean. Any values that randomly fell below zero were made to be zero intensity, mimicking the stochastic nature of the X-ray Free Electron Laser (XFEL) beam in Self-Amplified Spontaneous Emission (SASE) mode. The X-ray wavelength was also given a Gaussian distribution with RMS variation 0.5% about the mean of 1.5 Angstrom. The crystal was made to be 30 microns wide, and imperfections within it were simulated by breaking it up into 300 smaller crystals or "mosaic domains" Darwin (1922) that were miss-oriented relative to each other randomly using a top-hat distribution 0.5 degree in diameter. From shot to shot, the overall crystal orientation was also randomized to be equally likely in any direction. Background X-ray scattering, such as inelastic Compton scattering, elastic diffuse scattering from disorder in the crystal lattice, as well as 5 mm of air and 10 microns of liquid water were calculated with "nonBragg" using the equations described in the supplementary materials of Holton et al. (2014).

The sum of all these effects was taken as the expectation value (mean number) of X-ray photons falling on each pixel of the simulated X-ray detector, which was given 512 x 512 square pixels 172 microns wide and positioned 80 mm down-range from the crystal position. The expected mean number of photons on each pixel was fed through a Poisson distribution to obtain an "observed" number of photon hits, reflecting the random nature of X-ray photon arrivals. From here on pixel values were stored as unsigned integers and a single pixel level change was made to equal a single X-ray photon. The detector point-spread function of a fiber-coupled CCD X-ray detector was simulated as described by Holton et al. (2012), and each pixel was also given a Gaussian calibration error of RMS 4%, an additional "read-out noise" equivalent to RMS 3x the signal of a single photon hit, and an offset of 10 pixel units to keep the signal from going negative. Any counts that exceeded the 16-bit dynamic range of this simulated detector were clipped at 65025 and then the dynamic range was compressed to 8 bits by taking the square root of the photon count. This has the elegant property of placing the standard error of every pixel value to unity because the error in counting N photons is sqrt(N). The resulting 8-bit image was then stored in Portable Greymap format.

To enhance the speed of these calculations the Fraunhofer grating sinc function was replaced by a much quicker step function with the same full-width-at-half-max (FWHM) and volume (-tophat_-spots option in nanoBragg). This much faster calculation preserved the spot shape and intensity without calculating subsidiary maxima that are obscured by the background intensity in this case anyway. An additional speed enhancement was attained by calculating the scattering of an 0.1 micron wide crystal and scaling up the resulting intensity by a factor of 2.7e7 to match that of a 30

Table 8: Search space for the feature extractors and the RF/SVM classifiers hyperparameter search.

| | Hyperparameter | Type | Values | Default |
|---|---|---|---|---|
| SVM | C | Ordinal | 1 or $2^x$ for x in {-5, -3, ..., 13, 15} | 1 |
| | $\gamma$ | Ordinal | 0 or $2^x$ for x in {-15, -13, ..., 1, 3} | 0 |
| | Class weights | Categorical | None | None |
| | | | Balanced | |
| | | | [0.35, 0.35, 0.1, 0.1, 0.1] | |
| | | | [0.3, 0.3, 0.133, 0.133, 0.133] | |
| | | | [0.25, 0.25, 0.166, 0.166, 0.166] | |
| RF | Number of trees | Ordinal | {10, 100, 1000} | 10 |
| | Max features | Ordinal | {$\sqrt{features}$, 0.25, 0.5, 0.75} | $\sqrt{features}$ |
| | Max depth | Ordinal | {None, 2, 4, 6, 8, 10, 20} | None |
| | Class weights | Categorical | None | None |
| | | | Balanced | |
| | | | [0.35, 0.35, 0.1, 0.1, 0.1] | |
| | | | [0.3, 0.3, 0.133, 0.133, 0.133] | |
| | | | [0.25, 0.25, 0.166, 0.166, 0.166] | |
| GLCM | Distances | Categorical | Any combination of {1, 2, 4, 5, 8} | 5 |
| | Angles | Categorical | Any combination of {0, 45, 90, 135} | 0 |
| LBP | Points | Ordinal | {4, 8, 16, 24} | 24 |
| | Radius | Ordinal | {0, 1, 2, 3} | 3 |
| SIFT | Clusters | Ordinal | {10, 25, 50, 100, 250, 500, 1000, 5000} | 100 |

micron crystal. The reason for this size reduction and scale-up is because the spots from a perfect 30 micron crystal are very much smaller than a pixel, and in the simulation they are unlikely to land in the exact center of a pixel, leading to large aliasing errors. This can be alleviated by heavily over-sampling the pixels, but for our purposes equivalent results are obtained by reducing the crystal size down to the point where the spot size is roughly equal to that of a single pixel, and then only 2x over-sampling was needed for accurate capturing of the integrated spot intensities.

## B   SEARCH SPACE SVM AND RF WITH FEATURE EXTRACTORS

SVM and RF search spaces with the feature extractors are summarized in Table 8. The search space of the feature extractors includes the hyperparameters mentioned in Section 4.1. SVM search space includes the cost of misclassification parameter (C) and the $\gamma$ parameter for the RBF kernel. RF search space comprises the number of trees in the forest, the maximum number of features used by the trees to find the best split, and the maximum depth of trees. The search spaces for both classifiers also include a "class weight" hyperparameter that assigns different weights to the entries classes. In the class weights, *None* indicates all classes have the same weight, *Balanced* shows all classes are weighted according to their number of samples, and the value arrays mean the weight given to each class of DiffraNet (from *blank* to *strong*).

## C   DEEPFREAK SEARCH SPACE

Our search space for DeepFreak includes some topologies and learning hyperparameters (Tables 9 and 10). The topologies we have designed for DeepFreak seek to increase the network downsampling, reducing training times and improving accuracy. Likewise, we search for the mix of initial learning rate, momentum, weight decay, and batch size that maximizes accuracy.

## D   DEEPFREAK BEST CONFIGURATIONS

Table 11 shows the three best topologies found by BOHB for DeepFreak.

Table 9: Possible topology adaptations for ResNet.

| Name of Variant | Default | 1 | 2 | 3 | 4 | 5 | 6 |
|---|---|---|---|---|---|---|---|
| Number of Filters | 64 | 64 | 64 | 64 | 64 | 64 | 64 |
| 1st convolution size | 7 | 7 | 7 | 7 | 7 | 7 | 7 |
| 1st convolution stride | 2 | 2 | 2 | 2 | 4 | 2 | 2 |
| 1st pool size | 3 | 3 | 3 | 3 | 3 | 3 | 3 |
| 1st pool stride | 2 | 2 | 2 | 2 | 2 | 2 | 2 |
| 2nd pool size | 7 | 9 | 13 | 7 | 7 | 7 | 15 |
| 2nd pool stride | 1 | 2 | 2 | 1 | 1 | 2 | 2 |
| block strides | 1, 2, 2, 2 | 1, 2, 2, 2 | 1, 2, 2, 2 | 2, 2, 2, 2 | 1, 2, 2, 2 | 2, 2, 2, 2 | 1, 2, 2, 2 |
| **Name of Variant** | **7** | **8** | **9** | **10** | **11** | **12** | **13** |
| Number of Filters | 64 | 64 | 8 | 8 | 64 | 16 | 16 |
| 1st convolution size | 7 | 7 | 7 | 7 | 7 | 7 | 7 |
| 1st convolution stride | 2 | 2 | 2 | 2 | 2 | 2 | 2 |
| 1st pool size | 3 | 3 | 3 | 3 | 3 | 3 | 3 |
| 1st pool stride | 2 | 2 | 2 | 2 | 2 | 2 | 2 |
| 2nd pool size | 8 | 7 | 7 | 7 | 9 | 9 | 7 |
| 2nd pool stride | 1 | 1 | 1 | 2 | 2 | 2 | 2 |
| block strides | 2, 2, 2, 2 | 1, 2, 3, 3 | 1, 2, 2, 2 | 1, 2, 2, 2 | 1, 2, 2, 3 | 1, 2, 2, 2 | 1, 2, 2, 2 |

Table 10: Search space for DeepFreak hyperparameter search

| Hyperparameter | Type | Values | Default |
|---|---|---|---|
| Topology | Categorical | [1, 13] | 3 |
| Learning rate | Log Real | [1e-4, 10] | 0.1 |
| Momentum | Real | [0.5, 1] | 0.9 |
| Weight decay | Real | [0.00001, 0.00005] | 0.00001 |
| Batch size | Categorical | {4, 8, 16, 32, 64} | 8 |

# E    SVM AND RF CONFUSION MATRICES

Tables 12 and 13 show the confusion matrices for SVM and RF respectively.

# F    BEST CONFIGURATIONS FOR THE REAL DATASET

Tables 14 and 15 show the best configurations found for our classifiers for the raw real dataset. Tables 16 and 17 show the best configurations found for our classifiers for the preprocessed real dataset.

Table 11: Three best configurations found by BOHB.

| | Topology | Learning rate | Momentum | Weight Decay | Batch Size |
|---|---|---|---|---|---|
| A | 6 | 8.6673e-04 | 0.7770 | 2.5380e-05 | 1 |
| B | 8 | 8.4474e-04 | 0.56168 | 3.4855e-05 | 1 |
| C | 1 | 5.9409e-03 | 0.70739 | 3.0193e-05 | 2 |

Table 12: Confusion matrix of SVM for the test set.

|  |  | Predicted class | | | | | |
|---|---|---|---|---|---|---|---|
|  |  | blank | no-crystal | weak | good | strong | Recall (%) |
|  | blank | 2069 | 0 | 0 | 0 | 0 | 100 |
|  | no-crystal | 0 | 3266 | 1 | 0 | 1 | 99.93 |
| True class | weak | 1 | 160 | 3142 | 44 | 0 | 93.90 |
|  | good | 0 | 0 | 40 | 2377 | 27 | 97.26 |
|  | strong | 0 | 0 | 0 | 22 | 1450 | 98.51 |
| Precision (%) | | 100 | 95.33 | 98.71 | 97.3 | 98.11 |  |

Table 13: Confusion matrix of RF for the test set.

|  |  | Predicted class | | | | | |
|---|---|---|---|---|---|---|---|
|  |  | blank | no-crystal | weak | good | strong | Recall (%) |
|  | blank | 2069 | 0 | 0 | 0 | 0 | 100 |
|  | no-crystal | 0 | 3266 | 2 | 0 | 0 | 99.94 |
| True class | weak | 0 | 53 | 3254 | 39 | 0 | 97.25 |
|  | good | 0 | 0 | 45 | 2368 | 31 | 96.89 |
|  | strong | 0 | 0 | 0 | 25 | 1447 | 98.30 |
| Precision (%) | | 100 | 98.4 | 98.58 | 97.37 | 97.9 |  |

Table 14: Best configuration of RF (left) and SVM (right) for the raw real validation set.

| Hyperparameter | Values |
|---|---|
| LBP radius | 1 |
| LBP points | 16 |
| Max Depth | 20 |
| Max Features | 0.5 |
| Number of Trees | 10 |
| Class Weights | [0.35, 0.35, 0.1, 0.1, 0.1] |

| Hyperparameter | Values |
|---|---|
| LBP radius | 1 |
| LBP points | 24 |
| C | 1 |
| $\gamma$ | $2^{-9}$ |
| Class Weights | Balanced |

Table 15: Best configuration of DeepFreak for the raw real validation set.

| Hyperparameter | Value |
|---|---|
| Number of filters | 64 |
| 1st convolution kernel | 7 |
| 1st convolution stride | 2 |
| 1st pool size | 3 |
| 1st pool stride | 2 |
| 2nd pool size | 9 |
| 2nd pool stride | 2 |
| Number of blocks | 3, 4, 6, 3 |
| Block strides | 1, 2, 2, 3 |

| Hyperparameter | Value |
|---|---|
| Learning rate | $1.4542 \times 10^{-4}$ |
| Momentum | 0.99589 |
| Weight decay | $1.8555 \times 10^{-5}$ |
| Batch size | 1 |

Table 16: Best configuration of RF (left) and SVM (right) for the preprocessed real validation set.

| Hyperparameter | Values |
|---|---|
| LBP points | 2 |
| LBP radius | 24 |
| Max Depth | 4 |
| Max Features | 0.75 |
| Number of Trees | 100 |
| Class Weights | Balanced |

| Hyperparameter | Values |
|---|---|
| LBP points | 1 |
| LBP radius | 24 |
| C | $2^{15}$ |
| $\gamma$ | $2^{-7}$ |
| Class Weights | None |

Table 17: Best configuration of DeepFreak for the preprocessed real validation set.

| Hyperparameter | Value |
|---|---|
| Number of filters | 8 |
| 1st convolution kernel | 7 |
| 1st convolution stride | 2 |
| 1st pool size | 3 |
| 1st pool stride | 2 |
| 2nd pool size | 7 |
| 2nd pool stride | 2 |
| Number of blocks | 3, 4, 6, 3 |
| Block strides | 1, 2, 2, 2 |

| Hyperparameter | Value |
|---|---|
| Learning rate | $4.8311 \times 10^{-3}$ |
| Momentum | 0.88172 |
| Weight decay | $2.1462 \times 10^{-5}$ |
| Batch size | 8 |

