# OpenReview forum: "DiffraNet: Automatic Classification of Serial Crystallography Diffraction Patterns"
_ICLR.cc/2019/Conference_

### Official Review · AnonReviewer3 · 2018-10-24
**The paper described a new open synthetic dataset for serial crystallography and compared three approach to classify these data. The paper is well written, clear and contribute an original dataset. The significance of the work need to be justified further by making comparison to real data.**

**Rating:** 8
**Confidence:** 4

**Review:**

The paper describes a new open synthetic dataset for serial crystallography generated by a simulator. Three methods are proposed and implemented to demonstrate the classification of these diffraction images. The results from these methods are compared and clearly show the ones achieve high performance. The article structure is clear and is well written. The experiments are carried out in a professional way and statistical analysis is shown. It will be better if the authors can demonstrate how the models obtained from training the synthetic data perform in real scenario. Please also add some discussion on how good the synthetic data simulate the real data. Some image comparison between the synthetic data and real data should be analysed.

---

> ### Comment · AnonReviewer2 · 2018-11-03
> **I think that realistic tests are crucial**
>
> You state that it would be better to perform experiments on real scenarios. I would argue that this is crucial to be perfomed. And also it is required to see the influence of domain-shift. How better would be pre-training on this synthetic data compared to pre-training on imageNet, for example.

---

> > ### Author Response · Authors · 2018-11-21
> > **Response from the authors**
> >
> > The authors agree that we need to generate results on real data. We have generated a real dataset and used it to improve and evaluate our models which achieve ~94% accuracy on real images using this dataset. We refer to our response to the second reviewer for details on the dataset and the methodology used to achieve these results. Concerning the pretraining, we refer to our responses to the second reviewer for our reasoning behind this.
> >
> > The main difference between real and synthetic data is that currently available categorized real images are noisier and darker than the synthetic images. Real images also include a horizontal shadow in the middle created by the crystallography equipment (the beamstop shadow). We preprocess the real images to make them more similar to the synthetic dataset, i.e., we crop these images and multiply them by a constant factor so that the mean pixel value of the real images is close to that of the synthetic images.
> >
> > We added a visual comparison between real and synthetic images to Figure 2 in the paper, which shows samples of the categories in our synthetic dataset.

---

### Official Review · AnonReviewer2 · 2018-11-03
**Interesting application domain, but no novelity in terms of methods; For a synthetic dataset, I would like to see the impact to real test data**

**Rating:** 3
**Confidence:** 5

**Review:**

This paper introduces a purely synthetic dataset for crystallography diffraction patterns. For this very specific application domain, this might be a very welcome approach, however, I feel the paper is not strong enough for ICLR for two reasons:

1. The scope is too narrow for ICLR. Only a limited readership will be interested in this specific problem. Since the contribution is mainly on the dataset level and not on the methodological level, I suggest submitting such an article in venues more focused on the application domain. I can see no contribution which is general enough to be interesting for the broader readership. A new dataset might be of interest if it is a really challenging one where current approaches cannot yield high performance levels while it would be easy for domain experts to recognize.

2. The experiments are only on synthetic data. I agree that synthetic data in general can be very useful, if generated correctly (this has been shown in many works). For a substantial article contribution, one should, however, in general add much more exhaustive experiments. Besides analyzing the behavior on synthetic data, one should perform tests on real data and see the influence of, e.g., pre-training on this synthetic dataset. Furthermore, comparison to pre-training on other datasets should be performed.

---

> ### Author Response · Authors · 2018-11-21
> **Response from the authors**
>
> We thank you for the feedback. We address the points raised below.
>
> 1. The authors believe that DiffraNet is a challenging dataset. This statement is supported by the following: 1) High-resolution images (512x512), which make off the shelf tools not useful without domain-specific adaptation (our architecture adaptations can be found in Table 8). Simply downsampling the input image to the image size accepted by ResNet leads to poor performance (96.79% training accuracy and 72.08% validation accuracy); 2) Our best accuracy on the real dataset is 94%, suggesting that a great deal of improvement can be achieved; 3) DiffraNet will evolve in an increasingly challenging dataset over time by introducing new important structures like images with ice rings on top of the diffraction patterns and images with overlapping lattices. There is considerable overlap between these classes and our current diffraction classes, which makes classification more challenging.
>
> In addition, the methodology that we introduced, namely, the automated search of algorithm parameters (i.e., feature extractors) and, type of model and models hyperparameters (i.e., AutoML), highlights a common current problem in computer vision which is the automation of end-to-end algorithms including the downsampling (when the input image size is different from the ImageNet 224x224 resolution). The size of the ImageNet images can be considered a specific choice of embedding in an arbitrary Cartesian structure that cannot fit other image resolutions. The fundamental question to ask is "How do we downsample from a generic input image size such that we can leverage the well-validated representational power of off-the-shelf CNNs optimized for 224 x 224 Cartesian embeddings?". Our work on the DiffraNet microcosm is a first step to answer this question and we plan to build on top of our methodology to effectively tackle this problem.
>
> Finally, one of the ICLR topics is "applications in vision, [...], or any other field", which makes our paper compelling for ICLR.
>
>
> 2. The authors agree with the reviewer. We have generated a real dataset for this purpose with 457 images which will be released together with the synthetic dataset. The results on the real data hit ~94% accuracy with our models. To achieve these results, we split this dataset into validation (~80%, 366 images) and test sets (~20%, 91 images) and use AutoML as described in the paper to find the best configurations for each of our models based on their accuracy on the real validation set. We do not add any real image to the training set. We will add these results to the paper before the rebuttal period deadline.
>
> We argue, however, that pre-training our models with other datasets (e.g. ImageNet) may not improve performance on data drawn from a domain with fundamentally different imaging physics. Dunnmon et al. (2018), for instance, have shown that there is little to no contribution to using ImageNet pre-trained weights on datasets larger than 20,000 images drawn from X-ray radiography, i.e., using pre-trained weights from ImageNet did not improve performance on their chest radiographs dataset. Our dataset represents a similar scenario, and thus we do not expect a great deal of improvement from pretraining on a fundamentally different data distribution. Besides, DeepFreak uses an adapted architecture, which means there is no pretrained model readily available for us.
>
>
> Jared A. Dunnmon, Darvin Yi, Curtis P. Langlotz, Christopher Ré, Daniel L. Rubin, Matthew P. Lungren. "Assessment of convolutional neural networks for automated classification of chest radiographs." In Radiology, 2018.

---

> > ### Comment · AnonReviewer2 · 2018-11-27
> > **Thank you for the new experiments - This could be done more elaborated, however**
> >
> > I agree, that ICRL also allows for applications, but still, a significant methodological novelty is needed. I leave it to the PC chairs to decide if this contribution is significant enough.
> > About the experiments, you mention in your comments that you use the same crystal. What are the reasons for that? Why is it not mentioned and discussed in the paper? I see that there is a bias if always the same crystal is used. Actually, the results even reveal that you had to optimize the system again on 80% of the data. Much more investigation has to follow before be can really understand the significance of the dataset.

---

> > > ### Author Response · Authors · 2018-12-01
> > > **Response from the authors**
> > >
> > > We thank the reviewer for the additional comments and suggestions.
> > >
> > > Crystallography experiments do not combine different structures, data is always collected and analyzed one structure at a time. We maintain this field-specific common practice in our dataset, hence, our synthetic and real images were generated using a single structure each. As a side note, the authors would like to emphasize that we formed ahead of time a team of cross-domain researchers to ensure that the dataset would follow the standard practice of the crystallography community.
> > >
> > > Our results show that a model trained on diffraction from a synthetic crystal structure generalizes to a different real crystal structure. Our model achieves up to 94% accuracy on real images, using a purely synthetic training set. We note that there are two challenges in the real/synthetic experiments, namely,  the reality gap and the different structures. Our new CNN topology overcomes both simultaneously. We believe a small validation set with real data is a reasonable requirement for these results.
> > >
> > > We have included in the paper that DiffraNet’s synthetic data was generated with a single crystal structure. We will further clarify in the camera ready that generating data with a single structure is common practice in crystallography. We will also add that our real data was generated with a single, and different, structure.

---

### Official Review · AnonReviewer1 · 2018-11-03
**Well written and organized paper. Domain concepts presented fairly clearly. The approach and choices of classification algorithms is well articulated and results interesting. Good combination of known algorithms on a purposely built dataset (Originality questionable).**

**Rating:** 5
**Confidence:** 4

**Review:**

Contribution:
	- Using a known parameters crystallography simulator (X-ray beam, structure being analyzed, environment (crystalline or not)) built a dataset (called DiffraNet) of 25,000 512x512 grayscale labeled images of resulting diffraction images of various materials/structures (crystalline or not) .
	- carried various classification approaches of the dataset (labelled) images in two steps:
		- Feature extraction (Scale Invariant Feature Transform with the Bag-of-Visual-Words approach as local feature extractor, and the Gray-level Co-occurrence Matrix and Local Binary Patterns as global feature extractor) then
		- Classification of the diffraction images is carried with three approaches. Two using images described by extracted features (from the previous feature extraction step) coupled with either random Forests or Support Vector Machines and a third consisting in a Convolution Neural Network (CNN) topology named DeepFreak.
		- The images are classified according to the diffraction patterns they encompass into one of 5 classes: blank, no-crystal, weak, good and strong. The last three describing presence of a crystalline structure.
		- A fine tuning step of the various algorithms was carried using AutoML optimization tools.
All algorithms were off the shelf publicly available implementations and have previously been used for such domain applications (crystallography patterns).
The approach and choices of classification algorithms is well articulated and results interesting.

A few questions though:
•	In what way the diffraction images are ‘synthetic’? Aren’t they actual diffraction images but in a controlled known and controlled setting: set of parameters (beam, structure to analyze)?
o	More like a library of diffraction pattern images for various materials/structures.
•	How many structures were analyzed (Were there 25000 for the 25000 pattern images), one image each?
o	This is to understand  the representability of the samples (structures) analyzed regarding the possible structures (Hundreds of thousands as per paper’s 2.1 ) .
•	What variations for each of the setting variabilities (X Ray beam(flux, beam size, divergence, and bandpass), crystal properties (unit cell, number of cells, and a structure factor table), and the experimental parameters (sources of background noise, detector point-spread, and shadows)) were used?
o	This is to assess the size of the pattern space.
•	Were any real-life setting obtained pattern samples classified using DiffraNet dataset patterns’ fine-tuned classification algorithms?
o	This is to assess the generalization level of the DiffraNet dataset patterns’ fine-tuned classification algorithms to real-life obtained patterns (relates to the previously stated representability of the samples).
o	If not, your statement “ … we plan to add new images and new classes that are common place in serial crystallography” (in 6. Conclusions) would be an appreciated validation of general usability of your DiffraNet fine–tuned setting.
•	Were all the structures analyzed crystalline?
o	It’s stated in Figure 2 and Table 6 that 2 classes are either blank or no-crystal but is that a known fact (purposely chosen) or no pattern images for crystalline structures due to inadequate experimental settings to uncover the crystalline nature of the analyzed structure?
•	Were the pattern images pre-processed in any manner before being classified?


Nota: In table 6, use no-crystal class as in Figure 2 for consistency.

---

> ### Author Response · Authors · 2018-11-21
> **Response from the authors (part 1)**
>
> Thank you for the questions, we have expanded the explanation of the simulation procedure in the paper and added a complete description of the procedure as an appendix. We also address the questions directly below.
>
> - In what way the diffraction images are ‘synthetic’? Aren’t they actual diffraction images but in a controlled known and controlled setting: set of parameters (beam, structure to analyze)?
>
> The images are synthetic in the sense that they have been generated via the nanoBragg simulator instead of being recorded from real X-ray crystallography experiments.
>
> - How many structures were analyzed (Were there 25000 for the 25000 pattern images), one image each?
>
> We have simulated a single crystal, but in 25000 orientations with different diffraction parameters. The X-ray beam intensity varied widely, as did the volume of crystalline material in that beam relative to non-crystalline matter.  This wide dynamic range is a big factor in making this kind of data difficult to analyze.
>
> - What variations for each of the setting variabilities (X-Ray beam (flux, beam size, divergence, and bandpass), crystal properties (unit cell, number of cells, and a structure factor table), and the experimental parameters (sources of background noise, detector point-spread, and shadows)) were used?
> o This is to assess the size of the pattern space.
>
> We use air and water as sources of background noise and vary parameters of the X-ray beam. The beam intensity varies widely from shot-to-shot with a Gaussian distribution and the RMS fluctuation of the X-ray pulses was made to be equal to the mean. The X-ray wavelength was also given a Gaussian distribution with RMS variation. We also vary crystal parameters as detailed above. We do not vary the other parameters for this dataset. Many of these parameters are varied from shot-to-shot, which creates a diverse dataset. This means each image in our dataset is generated with a different parameter setting, and consequently, results in a different pattern.
>
>
> - Were any real-life setting obtained pattern samples classified using DiffraNet dataset patterns’ fine-tuned classification algorithms?
> o Were all the structures analyzed crystalline?
> o It’s stated in Figure 2 and Table 6 that 2 classes are either blank or no-crystal but is that a known fact (purposely chosen) or no pattern images for crystalline structures due to inadequate experimental settings to uncover the crystalline nature of the analyzed structure?
>
> Following the reviewers' suggestions we have been working on testing our models in real experimental images. We have generated a real dataset for this purpose with 457 images generated from the same crystal. Our final objective is to provide the real dataset together with the synthetic dataset. We currently split this dataset into validation (~80% of the dataset, for a total of 366 images) and test sets (~20%, 91 images) and use AutoML tools to find the best configurations for each of our models based on their accuracy on the real validation set. So far, we have hit ~94% accuracy using this methodology, we will add these results to the paper before the rebuttal deadline. Regarding the synthetic dataset, our dataset uses a single structure (Photosystem II), which is crystalline. Images with category ‘noxtal’ and ‘blank’ have zero contribution from the crystal lattice, anything larger than zero went into ‘weak’, ‘good’, or ‘strong’.
>
> - Were the pattern images pre-processed in any manner before being classified?
>
> Yes, we took the square root of the 16-bit images to make them 8-bit. We also rescale them to the [0,1] range. For DeepFreak, we subtract the mean as is usually done in CNNs. We have added this information to the paper.

---

> > ### Author Response · Authors · 2018-11-21
> > **Response from the authors (part 2)**
> >
> > - All algorithms were off the shelf publicly available implementations and have previously been used for such domain applications (crystallography patterns).
> >
> > While it is true that several components of our methodology are publicly available, the topology of DeepFreak is a domain-specific CNN architecture that can leverage additional information in the serial crystallography high-resolution image (512x512). The authors tailored the downsampling of ResNet-50 (image size 224x224) as shown in Table 8. We have found that simply downsampling the input image to the image size accepted by ResNet leads to poor performance (96.79% training accuracy and 72.08% validation accuracy). We will clarify this in the paper. In addition, this is the first time that a comprehensive set of models (SVM, RF and CNN) are studied in the context of serial crystallography applications. Our study shows that a nifty combination of feature detectors, and SVM and RF perform well on the synthetic dataset with potential implications on the inference speed of the classification (DeepFreak takes ~122 seconds to infer on the 12600 images of the test set, SVM takes ~2.18 seconds, and RF only ~0.25 seconds), which is an important factor for the real-time aspect of the crystallography deployment in the beamline.
> >
> > - In table 6, use no-crystal class as in Figure 2 for consistency.
> >
> > Thank you, we have changed the tables to ensure consistency.

---

### Meta-Review · Area_Chair1 · 2018-12-14
**Radically different scores**

**Confidence:** 4
**Recommendation:** Reject

**Metareview:**

Reviewer ratings varied radically (from a 3 to an 8). However, the reviewer rating the paper as 8 provided extremely little justification for their rating. The reviewers providing lower ratings gave more detailed reviews, and also engaged in  discussion with the authors. Ultimately neither decided to champion the paper, and therefore, I cannot recommend acceptance.